# Efficacy, Safety, and Challenges of CAR T-Cells in the Treatment of Solid Tumors

**DOI:** 10.3390/cancers14235983

**Published:** 2022-12-03

**Authors:** Qiuqiang Chen, Lingeng Lu, Wenxue Ma

**Affiliations:** 1Key Laboratory for Translational Medicine, The First Affiliated Hospital, Huzhou University School of Medicine, Huzhou 313000, China; 2Department of Chronic Disease Epidemiology, School of Medicine, Yale School of Public Health, New Haven, CT 06520, USA; 3Yale Cancer Center and Center for Biomedical Data Science, Yale University, 60 College Street, New Haven, CT 06520, USA; 4Sanford Stem Cell Clinical Center, Moores Cancer Center, Department of Medicine, University of California San Diego, La Jolla, CA 92093, USA

**Keywords:** CAR (chimeric antigen receptor), antigen, heterogeneity, efficacy, safety, T cell exhaustion, CRS (cytokine release syndrome), ICANS (immune effector cell-associated neurotoxicity syndrome), hematological malignancy, solid tumor

## Abstract

**Simple Summary:**

Chimeric antigen receptor T cells (CAR T-cells) are engineered T cells that target tumor-associated antigens. CAR T-cell therapy is a novel developed immunotherapy initially for destroying hematological malignancies. Its great success in clinical practice of hematological malignancies encourages oncologists and scientists to use CAR T-cells for the treatment of solid cancers. However, the efficacy of CAR T-cells in solid tumors is not as good as expected in hematological malignancies. In this review, we summarized the efficacy, safety, and challenges of CAR T-cell therapy in the clinical management of solid tumors. We also discussed the potential strategies currently applied to improve the efficacy and safety of CAR T-cell therapy in solid tumors, and finally prospected the future study direction for CAR T-cell therapy.

**Abstract:**

Immunotherapy has been the fifth pillar of cancer treatment in the past decade. Chimeric antigen receptor (CAR) T-cell therapy is a newly designed adoptive immunotherapy that is able to target and further eliminate cancer cells by engaging with MHC-independent tumor-antigens. CAR T-cell therapy has exhibited conspicuous clinical efficacy in hematological malignancies, but more than half of patients will relapse. Of note, the efficacy of CAR T-cell therapy has been even more disappointing in solid tumors. These challenges mainly include (1) the failures of CAR T-cells to treat highly heterogeneous solid tumors due to the difficulty in identifying unique tumor antigen targets, (2) the expression of target antigens in non-cancer cells, (3) the inability of CAR T-cells to effectively infiltrate solid tumors, (4) the short lifespan and lack of persistence of CAR T-cells, and (5) cytokine release syndrome and neurotoxicity. In combination with these characteristics, the ideal CAR T-cell therapy for solid tumors should maintain adequate T-cell response over a long term while sparing healthy tissues. This article reviewed the status, clinical application, efficacy, safety, and challenges of CAR T-cell therapies, as well as the latest progress of CAR T-cell therapies for solid tumors. In addition, the potential strategies to improve the efficacy of CAR T-cells and prevent side effects in solid tumors were also explored.

## 1. Introduction

According to the American Cancer Society (www.cancer.org (accessed on 18 November 2022)), cancer remains the second most common cause of deaths in the United States, just behind heart disease. As of 2022, it is estimated that there have been 1.9 million new cancer cases and 609,360 deaths in the United States, or about 1669 deaths every day [1].

In addition to the traditional cancer treatments, such as surgery, chemotherapy, radiation therapy, and targeted therapies (e.g., Imatinib/Gleevec, Trastuzumab/Herceptin, etc.), immunotherapy, which re-activates immune defenses against cancer cells, is a promising new approach and has emerged as the fifth pillar of cancer treatment in the last decade.

Cell-based immunotherapy is effective in cancers. Activated T cells recognize tumor antigens in the form of tumor antigenic peptide fragments that are presented and bound and bind to major histocompatibility complex class (MHC) molecules on the surface of tumor cells. However, the failure of T cells to eradicate cancer cells can result from the expression of immune checkpoint proteins on the surface of T cells, including different mechanisms such as T-cell exhaustion and immunosuppression [2]. Over the past decade, a number of immune checkpoint molecules including programmed cell death protein 1 (PD-1), cytotoxic T lymphocyte antigen-4 (CTLA-4), lymphocyte-activation gene 3 (LAG3), T cell immunoglobulin and mucin domain-containing protein 3 (TIM3), T cell immunoreceptor with Ig and ITIM domains (TIGIT), and B- and T-lymphocyte attenuator (BTLA, also known as CD272) have been identified and well-studied in cancer [2,3].

One of the upmost achievements in cancer immunotherapies in the last decade has undoubtedly been the introduction of T cell-targeted immunomodulators, the immune checkpoint inhibitors (ICIs), such as antibodies against CTLA-4 and PD-1 or programmed death ligand-1 (PD-L1) [4]. The first ICI antibody, Ipilimumab (Yervoy), was approved in 2011, followed by Pembrolizumab (Keytruda) and Nivolumab (Opdivo) in 2014. These ICIs have been widely used to treat a variety of cancer types, including head and neck, lung, kidney, bladder, lymphoma, and melanoma, etc. [2]. These ICI-based immunotherapies are not focused on in this review.

Chimeric antigen receptor (CAR) T cell (CAR T-cell)-based immunotherapy is another greatest achievement in cancer immunotherapies. CARs are generated by artificially fusing a tumor-specific antibody single-chain variable fragment (scFv) to the CD3ζ chain of the T cell receptor (TCR) via a transmembrane linker domain. The scFv specifically recognize specific antigens expressed on the cancer cell surface, or intracellular antigens if the scFv is expressed as intracellular antibody (intrabody) or delivered into the cells [5]. Thus, CAR is a combination of antibody-derived extracellular proteins, typically derived from the intracellular signaling module of an antibody and T-cell signaling proteins [6]. The fusion constructs are then transfected into autologous or allogeneic cytolytic T lymphocytes as CAR T-cells.

CAR T-cells recognize and target tumor antigens through the binding of CAR to tumor-associated antigen (TAA) or tumor-specific antigen (TSA) independent of the TCR-MHC/peptide interaction [6]. CAR T-cells have emerged as an effective novel cancer therapy for hematological malignancies [7].

The flow of the production, application and monitoring of CAR T-cells is summarized in Figure 1. CAR T-cell activity can be monitored with flow cytometric assay [8]. In clinical application, lymphodepletion is needed before the infusion of CAR T-cells to patients, so that the persistence of infused CAR T-cells can be effectively prolonged, and the effectiveness of tumor treatment can be improved [9].

The specificity of T cells against tumor cells is mediated by CAR proteins. At present, pan-B cell CD19 CAR T-cells have shown unprecedented response rates in treating hematological malignancies including refractory (R/R) B cell malignancies [10,11,12]. In 2017, autologous anti-CD19 CAR T-cells received the first regulatory approval from the US Food and Drug Administration (FDA) for the treatment of pediatric B-cell acute lymphoblastic leukemia (B-ALL), diffuse large B cell lymphoma (DLBCL), and, more recently, mantle cell lymphoma (MCL) [13].

T cells isolated from a cancer patient (for making autologous CAR T-cells), or other healthy donors (for making allogeneic CAR T-cells) are activated using artificial antigen-presenting cells (aAPCs), transfected with the CAR-encoding viral vector, and then expanded to large numbers in a bioreactor. After expansion, the cells are washed followed by the infusion back to the patient or concentrated and cryopreserved for future use. Activity of circulating CAR T-cells can be monitored with flow cytometry at different time points (e.g., 1 month, 3 months, 6 months, etc.) [14] by staining the antibodies of CD62L (circulating innate lymphoid cell precursors) [15], CD45RO (memory T cells), CD45RA (naïve T cells), CD4 (T helper cells), and CD8 (cytotoxic T lymphocytes). Subpopulations were defined as CD62L+ CD45RO+ central memory T cells (T_CM_), CD62L- CD45RO+ effector memory T cells (T_EM_), CD62L- CD45RA+ cells (T_EMRA_), and CD62L^+^ CD45RA^+^ naïve T cells [8]. Cytokines including intracellular IFN-γ etc. can also be measured by flow cytometry [16].

A decade ago, CD19-targeting CAR T-cells showed efficacy in patients with chronic lymphocytic leukemia (CLL) [17] and ALL [18]. However, the wide clinical application of CAR T-cells in both diseases has stalled. Patients with CLL did not respond to CD19 CAR T-cells as often as expected, at least in part because of challenges of producing CAR T-cell using autologous T cells from the patients with underlying diseases, or long term chemotherapy treatment, which can lead to lymphocytopenia [19]. Meanwhile, severe lymphocytopenia caused by chemotherapy is associated with reduced survival [20,21]. It is important to note that lymphocytopenia is a different concept from lymphodepletion. Lymphocytopenia is a disorder that lacks lymphocytes. In contrast, lymphodepletion is to purposely eradicate regulatory T-cells (Tregs) and other immunosuppressive cells through treatments and to make a room for the new CAR T-cells, thereby increasing CAR T-cell expansion prolonging persistence [21,22,23].

Due to the impressive results of CAR T-cells in hematological malignancies, many scientists in academia and industries have been attempting to extend CAR T- therapy to solid tumors. In recent years, the concept of CAR T-cells has been used in the development of other cell-based immunotherapies. According to the updated Cancer Cell Immunotherapy Pipeline 2022 from Cancer Research Institute, the number of active cell therapies developed in 2022 including CAR T-cells, TCR T-cells, NK and NKT cells, TIL cells, and tumor-associated antigen (TAA)/tumor-specific antigen (TSA) targeted T cells from preclinical to clinical trials and marketing is 2756, compared to 2031 in 2021. Of these, 1432 are active CAR T-cell therapies, 280 more than 1150 in 2021 [24]. Figure 2 illustrates the changes in the number of active CAR T-cell therapies in the past two years.

However, CAR T-cell therapy has been disappointing in the treatment of solid tumors and faces many challenges. Currently, a single antigen-targeted CAR T-cell therapy for solid tumors frequently fails, and no dual antigens-targeted CAR T-cell therapies have been approved for marketing yet in the world. To overcome tumor-defense mechanisms including immunosuppression (immunosuppressive cytokines secreted by solid tumors), antigen escape, and physical barriers to infiltrate into solid tumors, more sophisticated engineering approaches are required to develop effective CAR T-cell therapies.

This article reviewed the status of CAR T-cell therapy in the treatment of solid tumors (and also briefly summarized the treatment status of CAR T-cells in hematological malignancies), possible causes of failure, potential solutions, and the progress of CAR T-cell therapy for solid tumors and discusses the possible significance of CAR T-cell therapy for cancer treatment in the future.

## 2. Status of CAR T-Cell Therapy in Hematological Malignancies

CAR T-cell therapies have been approved for the treatment of the following types of cancer, including B-ALL in children and young adults up to age 25, DLBCL, primary mediastinal large B-cell lymphoma, large B-cell lymphoma transformed from follicular lymphoma, high grade B-cell lymphoma, aggressive B-cell lymphoma not otherwise specified (NOS), follicular lymphoma, mantle cell lymphoma, and multiple myeloma. CAR T-cell therapy is for patients who have tried two or more treatments, but their cancer has not improved or has relapsed. However, the efficacy of CAR T-cells in hematological malignancies is limited and has many challenges for its clinical applications [25]. The following paragraph is a quick overview on CAR T-cell therapy for hematological malignancies as the basis of CAR T-cell therapy for solid tumor.

Since 2017, the FDA has approved six CAR T-cell therapies for the treatment of CD19 or B-cell maturation antigen (BCMA, also referred as TNFRSF17, CD269) expressing hematological malignancies including ALL, non-Hodgkin lymphoma (NHL), and multiple myeloma (MM). Brexucabtagene (Tecartus) is the first CAR T-cell product approved for patients with MCL, a very aggressive disease with the worst prognosis among B-cell lymphoma and a poor response to chemotherapy. Other CAR T-cell products approved for the treatment of malignancies are tisagenlecleucel (Kymriah) for ALL, and axicabtagene ciloleucel (Yescarta) for DLBCL. BCMA is one of the most specific and highly expressed antigens on myeloma cells and offers a promising target in R/R MM [26]. BCMA as a target of CAR-T therapy has been approved for the treatment of MM. As of September of 2022, there are currently four approved CAR T-cell products available for NHL, two for B-ALL and one for MM, with more related products currently in the pipeline of clinical development for the malignancies. All the FDA-approved CAR T-cell product names, indications, manufacturers, approval dates, and one-time infusion costs are summarized in Table 1.

CD19 is an attractive target for cancer immunotherapy because it is expressed in the majority of B-cell malignancies, including 80% of ALLs, 88% of B-cell lymphomas, and 100% of B-cell leukemia, while its expression is very limited in normal B cells [27,28]. The “ideal” antigens currently approved for immunotherapy of hematological malignancies are CD19, CD20, and BCMA [27,28]. CAR T-cells eliminate tumor cells by directly targeting tumor cells expressing CD19 or CD20 or BCMA, thereby inducing a selective toxicity of the targeted tumor cells. A decade ago, anti-CD19 CAR T-cells first showed efficacy in patients with CLL and ALL. In addition, CD22 is also expressed on most B-ALL and is usually retained even after CD19 loss [29]. CD22 CAR T-cells have been shown to be effective in B-ALL patients who are not eligible for CD19 CAR T-cell therapy [30]. Unfortunately, most of the patients with B-ALL eventually relapse after CD22 CAR T-cell therapy [31] due to reduced or diminished CD22 expression levels in B-cell lymphoblasts (immature B cells). CD22 CAR T-cell therapy might still be a salvage therapy for B-ALL patients with an incomplete loss of CD22 expression (except a standard treatment). Nevertheless, a previous report showed that the second infusion of humanized CD22 CAR T-cells partially produced a suboptimal anti-leukemia response, with no expansion of CAR T-cells [30]. An ongoing clinical trial of autologous CD22 CAR T-cells for relapsed or R/R B-cell malignancies or ALL are conducted at Stanford Cancer Institute (Palo Alto, CA, USA). In addition, a Phase I study of CD22 CAR T-cells in patients with R/R hairy cell leukemia and its variant is ongoing at National Institutes of Health (NIH) Clinical Center (Bethesda, MD, USA). The participants would be followed closely for six months, and then less frequently for at least five years. CAR T-cell therapies targeting either CD19 or CD22 alone have potent antitumor effects, but antigen escape-mediated recurrence frequently occurs. Dual CAR targeting might be applied to overcome the issue [32,33]. Loss of target antigens [25,29,34,35] or poor CAR T-cell persistence [17,18] are primary factors limiting the efficacy of CAR T-cell therapies.

Among the current FDA-approved CAR T-cell products, none of them are used for the treatment of acute myeloid leukemia (AML). One of the challenges is no ideal antigens existing on AML cells. However, several potential target antigens (e.g., NKG2D ligands, C-type lectin-like molecule-1 (CLL-1), FMA-like tyrosine kinase 3 (FLT3), CD33, and CD23) are under investigation for AML treatment with CAR T-cell therapy [36]. Given that targeted antigens are usually shared between AML cells and myeloid progenitors, switchable CAR-T cells constitute a key strategy in the construction, thereby increasing safety. In addition, CD123, the transmembrane alpha chain of the IL-3 receptor, is strongly expressed on AML cells and is thought as a promising target. Khawanky et al. developed the third generation of anti-CD123 CAR T-cells with a humanized CSL362-based scFv and a CD28-OX40-CD3ζ intracellular signaling domain [37], and demonstrated anti-AML activity without affecting the healthy hematopoietic system, or causing epithelial tissue damage in a xenograft model of MOLM-13 [37].

A phase I clinical trial (NCT04318678) of anti-CD123 CAR T-cell therapy is under investigation for the treatment of AML/myelodysplastic syndrome (MDS), T- or B-ALL or blastic plasmacytoid dendritic cell neoplasm (BPDCN). The primary purpose was to find the maximum (highest) dose of CD123 CAR T-cells, at which it is safe to the patients. The side effects of the chemotherapy as well as the CD123 CAR T-cell product were also reported on the recipient’s body, and the primary endpoint included overall survival.

According to the latest data from Cancer Research Institute [24], the total number of active CAR T-cell therapies for leukemia conducted in 2022 are 134. Of these 134 therapies, 78 CAR T-cell products targeted CD19, 26 targeted BCMA, 8 targeted CD20, 11 targeted CD22 and CD123, respectively (Figure 3).

Durable remission after CAR T-cell therapy in hematological malignancies is not guaranteed. More than 50% of patients with B-cell ALL have been reported to relapse within 12 months after the treatment with CD19 or CD22 CAR T-cells [38,39]. Subgroup analysis of patients with B-cell malignancies showed that the ORR of patients with ALL, HL, NHL and CLL were 79% (95% CI: 70–86%), 37% (95% CI: 21–56%), 50% (95% CI: 23–78%) and 68% (95% CI: 45–84%), respectively [40]. Relapse rates are as high as 75% in patients with hematological malignancies treated with CD19 CAR T-cells [41]. One possibility is due to the increased contraction and transient persistence of CAR T-cells when exposed to antigen for a prolonged period [34].

Antigen loss or escape is another common cause of resistance to CD19-targeted immunotherapy [29,41]. Antigen variant (mutations and/or splicing variants in CD19 gene)-caused escape accounts for 7% to 25% relapse of patients treated with CD19-targeted CAR T-cells due to resistance of CAR T-cells [42]. CD19 CAR T-cell therapy also can lead to a deficient or low expression of CD19, which, in turn, result in resistance to the therapy, consequently bringing to DLBCL progression. Spiegel et al. reported that more than 50% of DLBCL patients treated with CD19 CAR T-cells experienced progressive disease because CD19 was absent or low in these patients [39].

According to recent studies, the global average success rates of CAR T-cell therapy is 50–80%. Since 2011, several large clinical trials have demonstrated that CD19 CAR T-cells have CR rates of 68% to 93% in patients with R/R B-ALL. Relapse remains common over time, occurring in about 40% to 50% of patients [43,44]. However, efficacy data are scarce on the patients with high-risk features including BCR-ABL^+^, TP53 mutation, extramedullary disease (including CNS leukemia), or relapse after transplantation [43].

A bi-specific CAR T-cell targeting CD19 and/or CD22 (CD19-22.BB.z-CAR) was developed to prevent disease recurrence in these patients after CD19 CAR T-cell therapy, and a Phase I clinical trial (NCT03233854) in patients with R/R B-ALL and DLBCL was conducted. The primary endpoint was feasibility and safety of manufacturing, and the secondary endpoint was efficacy. The results showed that B-ALL patients (n = 17) had 88% CR response rate, 29% CR for DLBCL patients (n = 21), while 50% (5 of 10) of the patients with B-ALL and 29% (4 of 14) of the patients with DLBCL relapsed because of CD19^deficiency/low^. However, these relapses were not associated with CD22^deficiency/low^ disease [39]. CD22 stimulation of CD19/22 CAR T-cell products also showed a reduced cytokine production when compared with CD19 alone [39]. Targets of CAR T-cell therapies for hematological malignancies include such as CD19, BCMA, CD22, CD20, CD123, TAA, CD33, CD30, CD38, and CS1.

CLL patients do not respond to CAR T-cell therapy as often as expected, in part due to the challenges of manufacturing products from patients whose T cells were unsuitable, due to either the underlying disease or exposure to years of chemotherapy. In patients with ALL, severe toxic effects led to delays in testing of modified dosing strategies, and even to completely discontinue trials. The toxic effects are the result of highly potent CAR T-cells, which leads to a significant cytokine elevation and increases blood–brain barrier (BBB) permeability and cerebral edema in a series of high-profile cases [45].

In addition to CD19 deficiency or low expression levels, the barriers of an effective CAR T-cell therapy also include severe life-threatening toxicity, such as cytokine release syndrome (CRS), the most common type of toxicity [46,47], modest anti-tumor activity, antigen escape, restricted CAR T-cell trafficking, and limited tumor infiltration [25].

## 3. CAR T-Cell Therapy in Solid Tumors

With the promising results of CAR T-cell therapy in the treatment of hematological malignancies, scientists have begun to extend CAR T-cell therapy to metastatic solid tumors, including lung, ovarian, breast, prostate, liver, kidney, stomach, pancreatic, and colon cancer. However, response rates of CAR-T therapy in patients with solid tumors much lower than those with hematological malignancies. The ORR of CAR T-cell therapy in patients with solid malignancies was 20% (95% CI: 11–34%) vs. 71% (95% CI: 62–79%) in those with hematological malignancies. Disappointing CAR T-cell therapies in the treatment of solid tumors [48,49] indicate the potential issues in solid tumors as follows: (1) TAA antigen identification (particularly tumor-specific antigens), expression level, and susceptibility to CAR T-cells, (2) tumor infiltration, CAR T-cells may not be able to penetrate tumor tissue through the vascular endothelium, and (3) survival of CAR T-cells in TME.

In 2010, a patient with colon cancer metastasis to the lung and liver died after ERBB2-targeting CAR T-cell therapy [50]. The potential cause of death may be that CAR T-cells recognize low levels of ERBB2 on lung epithelial cells, thereby triggering cytokine storms.

In 2021, Tmunity Therapeutics, a clinical-stage biotherapeutics company, halted the development of its lead CAR T-cell product after the deaths of two patients in a clinical trial. The patients reportedly died from immune effector cell-associated neurotoxicity syndrome (ICANS) (Carroll J. Exclusive: Carl June’s Tmunity encounters a lethal roadblock as 2 patient deaths derail lead trial, raise red flag forcing rethink of CAR T-cells for solid tumors. Endpoints News. 2 June 2021. Accessed on 3 June 2021. https://bit.ly/3wPYWm0).

The greatest challenge of CAR T-cell therapy for solid tumors is to find a tumor antigen that is uniquely expressed on the surface of solid tumor cells to provide CAR T-cell specific target. However, a major limitation of CAR T-cell therapy is that most proteins are tumor-associated antigens (TAAs), which are also expressed at low levels in normal cells, making it difficult for CAR T-cells to specifically target tumor cells without impairing healthy cells. Additionally, CAR T-cells are limited in trafficking to and infiltrating solid tumors as an immunosuppressive tumor microenvironment and physical tumor barrier [25]. Current CAR T-cell products for solid tumors are single-target CAR T-cells, dual-target CAR T-cell therapies have not yet been approved for marketing.

CAR T-cell therapy shows a strong clinical efficacy in hematological malignancies with a complete remission (CR) of around 30–40% in treating advanced B-cell malignancies [51], but not yet in solid tumors except for some individual cases [52]. According to 2022 AACR, Haanen and colleagues conducted a clinical trial to evaluate the early efficacy and safety of the CAR T-cell product targeting CLDN6. Among the 14 patients who were evaluable at six weeks after infusion, 4 patients with testicular cancer and 2 with ovarian cancer experienced a partial response (PR), with an overall response rate of nearly 43% (https://www.aacr.org/about-the-aacr/newsroom/news-releases/new-car-t-cell-therapy-for-solid-tumors-was-safe-and-showed-early-efficacy/#:~:text=Among%20the%20study%20participants%20who,at%2012%20weeks%20after%20infusion (accessed on 18 November 2022)).

According to the updated data from Cancer Research Institute [24], the total number of active CAR T-cell therapies for solid tumors conducted in 2022 was 27. Of these 27 therapies, 4 CAR T-cell products target HER2, 6 target MSLN, 7 target GPC2/3, and 10 target EGFR. These newly developed T-cell therapies in solid tumor are summarized in Figure 4.

In addition, according to the information from the website of the Clinical Trials (https://www.clinicaltrials.gov (accessed on 25 October 2022)), more CAR T-cells are under investigation in phase I clinical trials for solid tumors. The efficacy of CAR T-cells in solid tumors has not been supported. The following are some main CAR T-cell products developed for the treatment of solid tumor. Except a few of them are discussed in the text based on their attention and clinical accidents in clinical trials, others cannot be described here due to the spatial limitation, but they are summarized in Table 2.

### 3.1. CAR T-Cells Targeting Prostate TAA

Prostate-specific membrane antigen (PSMA) is a type II integral membrane glycoprotein highly expressed in prostate cancer and is a diagnostic and prognostic marker, which is a tumor-associated antigen (TAA). PSMA levels in prostate cancer are 100 to 1000 times higher than in normal tissues [53]. Meanwhile, high levels of PSMA are associated with the aggressiveness of human malignancies [54,55]. In addition, PSMA is also highly expressed in tumor neovascularization [56]. Increased PSMA expression is an independent predictor of prostate cancer recurrence. Therefore, PSMA becomes an attractive new therapeutic target with the most minimal tissue penetration for the development of anti-PSMA CAR T-cell therapy in prostate cancer [57]. Anti-PSMA CAR T-cells have robust killing ability against human prostate cancer cells and demonstrated strong expansion and cytotoxicity potential in prostate cancer cells [58]. Clinical trials conducted by Junghans et al. [59] and Slovin et al. [60] confirmed the safety and efficacy of PSMA-targeted CAR T-cells for prostate cancer. However, PSMA is also expressed in benign prostatic epithelial cells and normal prostate tissue.

One CAR T-cell product targeting PSMA, P-PSMA-101, is from Poseida Therapeutics (San Diego, CA, USA), which was designed to target prostate cancer cells expressing the cell-surface antigen PSMA (https://poseida.com/science/pipeline (1 December 2022)). In preclinical studies, P-PSMA-101 has been shown to eliminate tumor cells to undetectable levels in 100% of animals, with only one incidence of relapse in the lower dose (NCT04249947). Based on published literature, no other product candidate has shown complete elimination of solid tumors in this preclinical model. A phase I clinical trial of P-PSMA-101 CAR T-cells in patients with metastatic castration resistant prostate cancer (mCRPC) has been conducted and a dose escalation trial of P-PSMA-101 is ongoing.

Another CAR T-cell product targeting PSMA, TmPSA01, is from Tmunity Therapeutics (Philadelphia, PA, USA), which is a pioneer and lead CAR T-cell product, which is at the forefront of advancing CAR T-cell applications. Unfortunately, TmPSA01 has been halted after two patients died of neurotoxicity. This work identified potential barriers to CAR T-cell therapy for solid tumors. Notably, the researchers identified cases of ICANS. Details of the news were first reported by endpoints and confirmed by Tmunity. This news has profound implications for the broader push to toward cell therapy as a treatment for solid tumors (https://www.fiercebiotech.com/biotech/tmunity-stops-solid-tumor-car-t-trial-after-2-patients-die (accessed on 30 October 2022)). At Tmunity, the setback has led to the end of the CART-PSMA-TGFβRDN study and the start of work on subsequent candidates with improved safety profile. Tmunity is aiming to file an IND in the second half of the year.

In addition to PSMA, prostate stem cell antigen (PSCA), another TAA for prostate cancer, is a membrane glycoprotein predominantly expressed in prostate cancer, which is expressed in 94% (105/112) of primary prostate tumors and 100% (9/9) of bone metastases [61]. In vivo studies have showed that anti-PSCA monoclonal antibodies inhibit tumor growth and metastasis formation, making PSCA potentially useful in immunotherapy programs for the treatment of prostate cancer [62,63]. Although the expression of PSCA is upregulated in most of prostate cancers, its biological role in prostate cancer remains unclear.

### 3.2. CAR T-Cells Targeting MSLN (ATA2271)

Mesothelin (MSLN) is a cell-surface antigen associated with tumor invasion, which is strongly expressed in many solid tumor types, including mesothelioma, lung cancer, breast cancer, and pancreatic cancer [64]. MSLN has emerged as an important target in CAR T-cell therapy. Phase I clinical trials have shown that MSLN-targeted CAR T-cell therapy is safe, but its efficacy is very limited due to insufficient tumor infiltration and the persistence of CAR T-cells [65]. Lv et al. constructed MSLN CAR T-cells using MSLN scFv, CD3ζ, CD28, and DAP10 intracellular signaling domain (M28z10) to target MSLN. The results of in vitro experiments showed that M28z10 T cells exhibited strong cytotoxicity and cytokine-secreting ability to gastric cancer cells. The in vivo experimental results showed that M28z10 T cells could induce gastric cancer regression and prolong the mouse survival in different xenograft mouse models [66].

The expression rate of MSLN was various among pathological types of solid cancer (serous 97%, clear cell 83%, endometrioid 77%, mucinous 71%, carcinosarcoma 65%), pancreatic adenocarcinoma (ductal 75%, ampullary 81%), endometrial carcinoma (clear cell 71%, serous 57%, carcinosarcoma 50%, endometrioid 45%), malignant mesothelioma (69%) and lung adenocarcinoma (55%) [67]. The highest prevalence of positive MSLN was found in ovarian cancer. Kachala et al. reported that MSLN overexpression is a tumor aggressive marker and is associated with increased risk of recurrence and decreased overall survival (OS) [68]. MSLN CAR T-cell therapy has the potential to treat a variety of solid malignancies that are overexpressed MSLN [64]. Schoutrop et al. evaluated the efficacy of MSLN-directed CAR T-cell therapy in an orthotopic mouse model of ovarian cancer. The results showed that MSLN CAR T-cell therapy significantly prolonged survival, but sustained tumor control was not observed [69].

Malignant pleural mesothelioma (MPM) is a rare but highly aggressive malignancy with limited treatment options [70]. It is characterized by resistance to treatment and poor survival [71]. The median OS of patients with MPM after the first-line treatment with cisplatin and pemetrexed was only 13 to 16 months, which was prolonged to 18.8 months after an addition of bevacizumab, but at the cost of increased toxicity [72]. MPM treatment guidelines from the National Comprehensive Cancer Network (NCCN) include the use of ICIs as the second-line treatment. Although the expression levels of PD-L1 and tumor mutation burden (TMB) are very low in patients with MPM [73], the responses still occurred to PD-L1 blockade [70]. The scientists at Memorial Sloan Kettering Cancer Center (MSKCC) developed and conducted the first-in-human phase I study of a regional, autologous, MSLN-targeted CAR T-cell therapy [74]. The results showed that intrapleural administration of 0.3 to 60 M mesothelin-targeted CAR T-cells/kg was safe and well tolerated in 27 patients (25 with MPM, one with metastatic lung cancer, another with metastatic breast cancer), and CAR T-cells were detected in peripheral blood for >100 days in 39% of patients. The median OS of patients receiving CAR T-cell infusion was 23.9 months (83% 1-year OS rate). Eight patients had a stable condition for ≥6 months; two patients showed complete metabolic responses after positron emission tomography (PET) scanning [74].

In addition to the traditional therapies described above, MSLN-targeted CAR T-cell therapy for patients with advanced mesothelioma using next-generation PD1DNR and 1XX CAR technology is also being tested in clinical trials. ATA2271, a next-generation autologous CAR T-cell therapy targeting MSLN, manufactured by Atara Biotherapeutics, Inc. (San Francisco, CA, USA) is currently under clinical investigation in patients with MPM. According to Atara Biotherapeutics, ATA2271 targets hard-to-treat solid tumors using proprietary 1XX CAR signaling and intrinsic PD-1 checkpoint inhibition.

An ongoing Phase 1 dose-escalation trial of advanced mesothelioma has demonstrated the early safety and durability of armored CAR T-cells in patients. The preliminary results of the next generation autologous MSLN-targeted CAR T-cell ATA2271 were presented at 2021ESMO Immuno-Oncology Conference (9 December 2021). But on 18 February 2022, scientists at MSKCC notified the FDA of a fatal serious adverse event (SAE) associated with a patient treated with autologous CAR T-cells.

According to the press release issued by Atala Biotherapy on 28 February 2022 (https://investors.atarabio.com/news-events/press-releases/detail/265/atara-biotherapeutics-provides-update-on-ata2271-autologous (18 November 2022)): the first 6 patients enrolled in the two lowest dose groups received either 1 × 10^6^ cells/kg (patients 1–3) or 3 × 10^6^ cells/kg (patient 4–6) of ATA2271 intrapleural treatment. No dose-limiting toxicities have been reported in either cohort. The reported patient event was related to the first patient in a third, higher-dose cohort (6 × 10^6^ cells/kg). The temporary suspension of ATA2271 study enrollment does not affect the ongoing work to promote IND; ATA3271 is a separate, off-the-shelf, allogeneic ATA3271. ATA3219, Tabelecleucel (tab-cel), and ATA188 all utilize Atara’s allogeneic EBV T-cell platform, the safety and tolerability have been validated by clinical studies and experience in approximately 400 patients in various disease areas where CRS has not been observed to date.

The ongoing (from 30 September 2020 to September 2023) Phase 1 trial of MSLN-targeted CAR T-cell therapy in patients with mesothelioma is sponsored by MSKCC (https://clinicaltrials.gov/ct2/show/NCT04577326 (18 November 2022)). Intrapleural administration of ATA2271 was well-tolerated at the lowest dose levels, and no CAR T-cell related adverse events (AEs) of Grade > 2 observed and no AEs of Grade > 3 have been observed in the study to date. All four patients had received at least four prior lines of therapy. Importantly, ATA2271 CAR T-cells persisted in peripheral blood of patients for more than 4 weeks and were associated with upregulated effector cytokines.

### 3.3. Other Tumor Antigens Used for CAR T-Cells in Solid Tumors

#### 3.3.1. MUC1

MUC1 is a transmembrane glycoprotein that is aberrantly glycosylated and overexpressed in a variety of epithelial cancers. Previous studies have confirmed that MUC1 is overexpressed in NSCLC tissues [75,76], and in about 70% of ovarian cancer [77]. Tumor-associated MUC1 (tMUC1) is different from the MUC1 expressed in normal cells and can be used as a biomarker and therapeutic target of cancer [78]. Zhou et al. reported that monoclonal antibody TAB004 specifically recognizes tMUC1 in all subtypes of breast cancer, including 95% of triple-negative breast cancer (TNBC), while retaining recognition of MUC1 in normal tissue [79]. The team transduced human T cells with MUC28z, a CAR comprised of the scFv of TAB004 coupled to CD28 and CD3ζ. The results showed that MUC28z was well expressed on the surface of engineered activated human T cells. MUC28z CAR T-cells showed significant target-specific cytotoxicity against a group of human TNBC cells.

#### 3.3.2. ICAM1

Intercellular adhesion molecule-1 (ICAM1) is a cell surface transmembrane glycoprotein receptor. ICAM1 has been reported to be overexpressed in lung cancer, pancreatic cancer [80] and renal cell cancer [81]. High levels of ICAM1 were correlated with metastasis and poor prognosis in cancer patients [80]. ICAM1 expression is increased in TNBC patients and can be up to 200-fold increase in lung metastases of TNBC patients [82]. The authors demonstrated ICAM1 generated a phage-displayed scFv library using splenocytes from ICAM1-immunized mice and selected a novel ICAM1-specific scFv, mG2-scFv. Using mG2-scFv as the extracellular antigen binding domain, the team constructed ICAM1-specific CAR T-cells and demonstrated potent and specific killing of TNBC cell lines in vitro and in vivo [83].

#### 3.3.3. EGFR

Epidermal growth factor receptor (EGFR) is a transmembrane protein involved in cell growth and differentiation. EGFR is overexpressed in a wide range of solid tumor types [84], it is critical to control the growth and survival of epithelial cells, including NSCLC [85]. EGFR targeted therapies includes tyrosine kinase inhibitors (TKIs, e.g., gefitinib and erlotinib, afatinib, Osimertinib) [86], phosphatidylinositol 3-kinase (PI3K) inhibitors, and antisense gene therapy. These EGFR TKIs have effectively replaced chemotherapy as the first line treatment [87], Unfortunately, EGFR is increasingly recognized as a biomarker of tumor resistance [84], since all patients with metastatic lung who initially benefit from EGFR-targeted therapies eventually developed resistance [88]. EGFR-specific CAR T-cells have been reported not only to trigger cell lysis of EGFR-positive TNBC in vitro, but also to inhibit the growth of mouse cell lines and patient-derived xenograft (PDX) TNBC tumors [89]. EGFR is also a target of immunotherapy [90,91]. EGFR monoclonal antibodies (mAbs, e.g., cetuximab, panitumumab, nimotuzumab, and necitumumab) have been developed for the treatment of cancer [92].

Li et al. showed that proliferation and anticancer effects of EGFR CAR T-cells in vitro depend on time (24 to 72 h) and antigen (with and without EGFR antigen stimulation), and the regression of EGFR-positive human lung cancer xenografts in vivo [93]. A phase I clinical trial of EGFR CAR T-cells (NCT03182816) demonstrated that EGFR CAR T- cell therapy was well tolerated in all nine patients in treatment of EGFR-positive advanced R/R NSCLC patients. The results showed that EGFR CAR T-cells were detectable in peripheral blood of eight patients, partial response (PR) was observed in one patient, stable disease (SD) in six patients, and progressive disease (PD) in two patients. The progression-free survival (PFS) of these 9 patients was 7.13 months (95% CI 2.71–17.10 months), and the median OS was 15.63 months (95% CI 8.82–22.03 months) [94].

#### 3.3.4. ROR1

Receptor tyrosine kinase-like orphan receptor 1 (ROR1), a member of ROR family, is a protein encoded by ROR1 gene, and it is overexpressed in cancer [95]. For example, 28.6% was found in 56 histologically confirmed lung adenocarcinoma (using a cut-off of 1), or in 51.8% of the cases using the median value as threshold [95]. It was reported that ROR1 repression inhibits the growth of lung adenocarcinoma regardless of EGFR status, and leads to multiple acquired resistance mechanisms, including EGFR T790M, MET amplification and hepatocyte growth factor (HGF) overexpression [96]. ROR1 CAR T-cells can effectively kill lung cancer cells in a three-dimensional tumor model of NSCLC. Wallstabe et al. reported that ROR1 CART-cell treatment not only showed strong antitumor activity in human lung cancer cell line (A549), but also infiltrate into cancer tissue and eradicated multiple layers of tumor cells [97]. This result provides a new strategy for the clinical treatment of lung cancer. A clinical trial (NCT02706392) has been conducted to evaluate autologous ROR1 CAR T-cells in patients with advanced ROR1-positive and stage IV NSCLC; the results have not been released yet.

#### 3.3.5. Trop2

Trophoblast cell surface antigen 2 (Trop2) is a widely expressed glycoprotein and a member of the epithelial cell adhesion molecule (EpCAM) family in many normal tissues, and overexpressed in a variety of human cancers, including gastric cancer [98] and breast cancer [99]. Trop2 has potential in promoting epithelial-mesenchymal transition (EMT) in human breast cancer [100]. Overexpression of Trop2 has prognostic significance [101]. A study demonstrated that intra-tumoral injection of bi-specific Trop2/PD-L1 CAR T-cells can significantly reduce the growth of gastric cancer, and the inhibitory effect is stronger than specific Trop2 CAR T-cells [102]. These results suggest that novel Trop2/PD-L1 CAR T-cells are involved in Trop2/PD-L1 and checkpoint blockade in gastric cancer, thereby promoting the cytotoxicity of CAR T-cells in gastric cancer and other types of solid tumors [102].

#### 3.3.6. TAG72

Tumor-associated glycoprotein 72 (TAG-72) is a pan-adenocarcinoma oncofetal antigen that is highly expressed in ovarian cancers, and increased expression is associated with disease progression. The recurrence of ovarian cancer after surgery and multidrug chemotherapy is frequent, and novel therapeutic methods are urgently needed [103]. TAG72 has been used as a target for CAR T-cell therapy. Humanized TAG72-specific CAR T-cells have been reported to show potential cytotoxicity and cytokine production in ovarian cancer. On the other hand, TAG72-based CAR T cells significantly reduced the proliferative potential and improved the survival rate of mice [104]. Shu et al. demonstrated that the co-expression of the TAG-72 CAR and the CD47-truncated monomer CAR on T cells (dual CAR T-cell strategy) may be effective in ovarian cancer, and applicable to other adenocarcinomas [105].

#### 3.3.7. CA9

Carbonic anhydrase IX (CA9/ or CA IX) is an enzyme encoded by the human *CA9* gene [106]. CA IX is overexpressed in many types of cancer, including clear cell renal cell carcinoma (RCC) [107,108,109], cervical cancer [110], and breast and lung cancer, and CA IX promotes tumor growth by enhancing tumor acidosis [111] as other CA family members [112]. CAIX is a highly expressed on the surface of tumor cells in RCC [107]; thus, CAIX is a potential therapeutic target. CAIX overexpression increased the expression of 6-Phosphofructo-2-Kinase/Fructose-2, 6-Biphosphatase 4 (PFKFB4) and EMT, and promoted the migration of cervical cancer cells. CAIX can promote metastasis of cervical cancer cells, thus its inhibitory effect can be used as a therapeutic strategy for cervical cancer [110]. Li et al. reported that CAIX CAR T-cells combined with sunitinib induced an effective antitumor response in an experimental model of metastatic RCC [113].

#### 3.3.8. CD133

CD133 (Prominin 1, PROM1) is a transmembrane protein whose mRNA and glycosylated forms are highly expressed in a variety of human cancer cells. CD133 is a cancer stem cell (CSC) marker and is associated with cancer progression and patient prognosis [114,115], including pancreatic cancer [116], colorectal cancer [117] and breast cancer [118,119]. Besides the application of CD133-targeted CAR T-cells in MLL leukemia [120], anti-CD133 CAR T-cells have been reported in a phase I trial including 14 patients with hepatocellular carcinoma (HCC), 7 patients with pancreatic carcinomas, and 2 patients with colorectal cancer [121]. The results demonstrated the feasibility, controllable toxicities, and efficacy of anti-CD133 CAR T-cell therapy, with 3 patients achieving PR and 14 patients achieving SD among the 23 patients enrolled.

#### 3.3.9. Integrin αvβ6

Integrin αvβ6 is an exciting biomarker and therapeutic target for pancreatic cancer, and it is highly expressed in almost 100% of pancreatic ductal adenocarcinoma (PDAC) cases [122]. It was reported that CAR T-cells expressing CXCR2 exhibited stronger anti-tumor activity against pancreatic tumor xenografts known to express αvβ6 [123].

## 4. Lesson on Safety of CAR T-Cells in Solid Tumors from Hematologic Malignancies

Experience in the safety of CAR-T therapy in hematologic malignancies has accumulated profound lessons, from which physicians may learn to guide their clinical practice in patients with solid tumors, and better manage the safety issues of CAR-T therapy.

CAR T-cell therapy may have some common mild side effects, including high fever and chills, dyspnea, severe nausea, vomiting, and/or diarrhea, dizziness or lightheadedness, headaches, tachycardia, fatigue, and muscle and/or joint pain. Some serious side effects may exist, including high levels of CRS and neurotoxicity (immune effector cell-associated neurotoxicity syndrome, ICANS), which make the doctors as walk on thin ice when prescribing CAR T-cell therapies. In addition to above side effects, CAR T-cell therapy for solid tumors also faces other safety risks, such as fatal macrophage activation syndrome (MAS) [124] and uveitis [125], etc.

CRS, neurologic symptoms (NS) and tumor lysis syndrome (TLS) are the common side effects caused by CAR T-cells. Of grade 3 or 4 adverse events, CRS accounts for 22%, neurologic events 12%, cytopenia lasting more than 28 days 32%, infections 20%, and febrile neutropenia 14%. Three patients died within 30 days of infusion due to disease progression. No deaths were attributed to tisagenlecleucel, CRS, or cerebral edema [126].

Despite potentially life-threatening toxicities, the benefits of CAR T-cell therapy far outweigh the risks, especially as these toxicities are being mitigated with increased experience and improved supportive therapies [42]. In clinical practice, safe use of CAR T-cells is both a skill and art. Learning to balance the efficacy and safety of CAR T cells in cancer treatment is critical [127]. Hopefully, one day, CAR T-cells will have both high efficacy and high safety (Figure 5).

### 4.1. Cytokine-Release Syndrome (CRS)

CRS is the most common adverse effect after CAR T-cell infusion. CRS toxicity usually occurs within the first week after CAR T-cell therapy, and typically peaks within 1–2 weeks of cell administration [128]. CRS is an acute systemic inflammatory syndrome characterized by high fever and chills, as well as multiple organ dysfunction (difficult breathing, severe nausea, vomiting, and/or diarrhea, feeling dizzy or lightheaded, headaches, fast heartbeat, feeling very tired, muscle and/or joint pain). This is caused by the release of cytokines into the body following the activation of immune cells (especially T/CAR T-cells) during immunotherapy [127]. CAR T-cell therapy-associated CRS remains a major hurdle before its widespread use. CRS can be fatal if not properly identified and managed. In addition, neurotoxicity, called CAR T-cell-related encephalopathy syndrome (CRES), is the second most-common adverse event and can occur concurrently with or after CRS. The management of CRS is a big concern in these indications. IL-6 is identified as a key risk factor of CRS, which is a potential target in developing strategies to improve safety.

### 4.2. Immune Effector Cell-Associated Neurotoxicity Syndrome (ICANS)

ICANS is another common and unique toxicity after CAR T-cell therapy, occurring in up to 67% of leukemia patients and 62% of lymphoma patients [129]. Particularly in patients with ALL, this complication, however, did not prevent the product from entering the market of hematological malignancies. It has been reported that the level of neurotoxicity in patients with severe neurotoxicity is associated with the disruption of the blood-cerebrospinal fluid (CSF) barrier, but not with the white blood cell count or CAR T-cell number in CSF [129].

The mechanism of ICANS is relatively poor understood. Tmunity believes that if this adverse event is going to be a problem of CAR T-cell products in solid tumors, better understanding of the mechanism is a priority to bypass the barrier. The patients reportedly died from immune effector cell-associated ICANS in CAR T-cell therapies. “What we are discovering is that the cytokine profiles we see in solid tumors are completely different from hematological cancers”, Usman “Oz” Azam, the former president and CEO of Tmunity mentioned in an interview with Endpoints News (https://www.onclive.com/view/car-t-cell-therapy-trial-in-solid-tumors-halted-following-2-patient-deaths (accessed on 18 October 2022)).

### 4.3. Macrophage Activation Syndrome (MAS)

Macrophage activation syndrome (MAS) is a severe life-threatening complication related to hemophagocytic lymphohistiocytosis (HLH). It is characterized by the uncontrolled activation and proliferation of T lymphocytes and macrophages, resulting in the release of high levels of inflammatory cytokines [130]. HLH can be divided into primary HLH (pHLH) and secondary HLH (sHLH). The former is caused by an inherited disease, such as severe systemic lupus erythematosus (SLE has at least a partial genetic component), the latter is caused by other diseases including infections, malignancy, and autoimmune diseases [131]. The clinical syndrome of HLH include fever, hepatosplenomegaly, abnormal liver function, decreased blood cells, increased triglycerides, serum ferritin, and decreased fibrinogen. Systemic inflammatory response rarely evolves into a fulminant hemophagocytic lymphohistiocytosis (HLS) and MAS, which are associated with a high mortality rate. HLH/MAS caused by CAR T-cell therapy is an unusual manifestation of CRS with immune-mediated multi-organ failure, poor prognosis, and a challenging diagnosis. CAR T-cell therapy related HLH/MAS has a distinct malignancy-related PET-CT scan, showing a paradoxical response of hyper-inflammation in CAR-T therapy-related HLH/MAS patients [132]. Consistently, flow cytometry results showed the expansion of CAR T-cell existing in peripheral blood (PB), and the increased CAR T-cells at different follow-up time points [132]. Anti-IL-6 therapy, steroids, anakinra (a recombinant IL-1 receptor antagonist) and emapalumab (an anti-IFNγ, approved by the FDA) are recommended for the management of HLH/MAS [128,132].

## 5. Challenges of CAR T-Cell Therapy in Solid Tumors

CAR T-cell therapy has revolutionized the treatment of hematological malignancies, but its use in solid tumors has been challenging. The greatest challenges in generating CAR T-cells for the treatment of solid tumors include (1) cell recognition: solid tumors exhibit a considerable degree of antigen heterogeneity, with only a subpopulation of the cells expressing the target antigen. Most proteins on solid tumor cells can be targeted, but they are also expressed even at very low levels on normal cells, making it difficult for CAR T-cells to specifically target tumor cells without jeopardizing healthy cells. In addition, antigen expression levels on various tumor cells may impair CAR T-cell function because the diversity of antigens makes it difficult to identify TSA. CAR-Ts can effectively redirect CTLs to surface antigens that are highly expressed on tumor cells. However, the low expression of several TAAs on normal tissues hinders their safe target by CAR T-cells due to on/off-target tumor effects [133]. (2) Cell trafficking: CAR T-cells are required as cytotoxic CD8+ T cells to home into malignant sites after infusion, navigate the complicated TME, form efficient interactions with cancer cells, deliver their cytotoxic activities, and ultimately persist [134]. Yet, unlike in treating hematological malignancies, CAR T-cell therapy is more limited in solid tumors because CAR T-cells may not be able to infiltrate into solid tumor tissues through the vascular vessels [135]. The host and TME interactions with CAR T-cells critically alter CAR T-cell function [25]; (3) cell surviving: TME is widely considered to be detrimental to T cells, and CAR T-cells have limited activities in solid tumors. The glycolytic metabolism of tumor cells makes the environment hypoxic, acidic, and nutrient-deficient, which is easy to produce oxidative stress, thereby affecting IL-2 signaling and T cell proliferation [136]. Additionally, in an inflammatory environment, tumor cells express ligands (e.g., Gal9 and PD-L1, etc.) that bind to the T cell inhibitory receptors TIM-3 and PD-1, respectively, further promoting T cell exhaustion [137,138,139]. In addition, continuous exposure to a CD19 × CD3 bispecific molecule induces T-cell exhaustion [140].

CAR T-cell exhaustion is a major limitation to their efficacy especially in solid tumors. T cell exhaustion is a state of T-cell dysfunction characterized by a progressive loss of effector function during neoplastic disease. Continuous tumor antigen stimulation, immunosuppressive TME, alteration of T cell-associated transcription factors, and metabolic factors all can result in T cell exhaustion [41]. Other factors, such as CAR T manufacturing (the structure and qualitative characteristics of CAR T structures), changes in the TME, previous treatments, or effects of neighboring cells can also lead to CAR T-cell exhaustion and affect response outcomes. These exhausted CAR T-cells have no proliferative capacity and lose the ability to produce IFNγ, chemokines, and degranulation [141]. As a result, they lose the ability to eliminate tumor cells. Furthermore, most CAR T-cells are primarily autologous, and CAR T-cell therapy has significant deficiencies with T-cell exhaustion potential. In summary, CAR T-cell exhaustion is believed to be due to sustained antigenic stimulation, as well as an immunosuppression of TME, maintaining CAR T-cell effector function, sustaining, and achieving clinical potency remains a critical challenge [41].

Moreover, other challenges also exist, which limit the therapeutic efficacy of CAR T-cells in solid tumors. For example, the manufacture of low-quality CAR T-cells, antigen escape (tumor cells of a significant portion of patients treated with these CAR T-cells display either partial or complete loss of target antigen expression) [25], and severe life-threatening toxicities. The other potential reasons for nondurable response to CAR T-cell therapy are summarized in Figure 6.

## 6. Improve the Effectiveness and Safety of CAR T-Cells

Single-cell RNA sequencing may accelerate the understanding of CAR T-cell therapy effectiveness and safety. Bai et al. used single-cell RNA sequencing and proteomics approaches to analyze the mechanism of resistance in ALL patients treated with CD19-targeted CAR T-cells. The authors presented 101,326 single-cell transcriptomes and surface protein profiles from the infusion products of 12 ALL patients, the results showed significant heterogeneity in antigen-specific activation states, with a deficiency of T helper 2 function associated with CD19-positive relapse vs. durable responders (remission, >54 months) [142]. These molecular mechanisms can be potentially used to boost specific the function of specific T cells to maintain a long-term remission.

To date, most clinical studies have focused on patients who respond to CAR T-cell therapies only but ignoring the 50% to 60% of patients who fail CAR T-cell therapy and relapse [143]. Future clinical trials for these patients need to be designed to develop optimal treatment strategies for these patients.

One possibility for patients who do not respond to CAR T-cell therapy is that many auto-CAR T patients received three or more courses of chemo treatment before cell collection, so that the collected T cells for CAR T were less suitable at baseline [144].

Unlike in hematological malignancies, another possibility for CAR T-cell therapy in solid tumors is limited by insufficient tumor infiltration, T cell dysfunction and exhaustion. Local delivery of CAR T cells in patients with solid tumors is a safe and feasible strategy, which can increase the proliferation and penetration depth of CAR T-cells in tumors, enhance the help of CD4, immune balance, and more metastasis to the metastasis site and drainage to lymph nodes [145]. In addition, cancer cells often lose antigen expression due to the inhibition of their antigen mRNA translations, thereby escaping antitumor immune surveillance and attack by T cells, including CAR T-cells [146]. Moreover, the immune evasion may also result from CAR T-cell dysfunction, unfavorable TME, or drug-resistant cancer cells [147].

The favored antigens of CAR T-cell therapy for solid tumors usually include TAA, HER2, MSLN, GD2, EGFR, GPC2/3, NY-ESSO-1, MUC1, PSMA, EBV, Claudin 18.2, etc. Cadherin 17 (CDH17) is a novel oncogene, biomarker, and attractive therapeutic target for the aggressive malignancies. Feng et al. demonstrated that anti-CDH17 CAR T-cells not only eradicate CDH17-expressing neuroendocrine tumors (NETs), gastric, pancreatic, and colorectal cancers in xenograft or autochthonous mouse models, but also do not attack normal intestinal epithelial cells, which also express CDH17 to cause toxicity [148].

Multi-target CAR T-cells is another strategy to improve the effectiveness of CAR T-cells in solid tumors. Combined targeting of two or more tumor antigens can offset antigen escape, thereby enhancing T-cell effector functions. This idea is derived from dual-target CAR T-cells in hematological malignancies. Trivalent CAR T-cells that co-target HER2, IL13Rα2, and EphA2 can overcome antigenic variation among patients with glioblastoma [149]. Currently, no dual-target or multi-target CAR T-cell products have been approved for marketing yet.

### 6.1. CD19 × CD22 CAR T-Cells

CAR T-cells targeting either CD19 or CD22 have shown remarkable activity in B-ALL. Research results showed that some patients with B-cell tumors who received CD19 or CD22 CAR T-cells still experienced disease progression and recurrence. The major cause of treatment failure is antigen downregulation or loss. CD19 and CD22 are all specially expressed on B cell malignancies. Current investigations of dual targeting antigen receptors have demonstrated encouraging results, providing a high degree of optimism that the efficacy and the broader application of CAR T-cell therapy will gradually increase in B-ALL treatment [42]. A phase I trial in pediatric and young adult patients with R/R B-ALL (n = 15) tested autologous CAR T-cells expressing both anti-CD19 and anti-CD22 and showed a remission rate of 86% at a month after treatment with a favorable safety profile. The 1-year OS and event-free survival (EFS) rates were 60% and 32%, respectively [150].

### 6.2. CD19 × BCMA CAR T-Cells

CD19 targeted CAR T-cell therapies have been successfully used in patients with B-cell hematological malignancies, and also demonstrated consistently high antitumor efficacy in the relapsed B-ALL, CLL and B-cell non-Hodgkin lymphoma (B-NHL) [151]. B-cell maturation antigen (BCMA), also known as tumor necrosis factor receptor superfamily (TNFRSF) member 17, is preferentially expressed in mature B lymphocytes, and its overexpression and activation are associated with MM [152]. The potential of BCMA-targeted therapies to improve the treatment landscape for MM has been outlined not only in preclinical models, but also the clinical data [152]. Clinical data indicated that the combination of anti-BCMA and anti-CD19 CAR T-cell therapies induced high response rates in patients with R/R MM. Further clinical trials with a median follow-up time of 21.3 months have demonstrated that the combination therapy of anti-BCMA with anti-CD19 CAR T-cells induced durable responses in patients with R/R MM, with a median response duration of 20.3 months and a median PFS of 18.3 months. In addition, this dual CAR T-cell treatment was associated with a manageable long-term safety profile [153].

### 6.3. CD33 × CLL1 CAR T-Cells

AML is morphologically characterized by heterogeneous leukemia cells from myeloblasts to differentiated myeloid elements [154]. Heterogeneous cells in AML can consequently offset killing effect of single target–based CAR T-cell therapy, resulting in disease relapse. CD33 is widely expressed in AML [155], and C-type lectin-like molecule-1 (CLL1) is highly expressed on AML leukemia stem cells (LSC) and blasts, but not on normal hematopoietic stem cells (HSC) [156]. Targeting both CD33 and CLL1 surface antigens may offer two distinct benefits. Simultaneously targeting both bulk leukemia cells and LSC not only comprehensively ablates AML disease, but also overcome the defect of single antigen loss, thereby preventing relapse. A Phase I study has evaluated the safety and effectiveness of this CD33-CLL1 dual CAR T-cell therapy in R/R AML patients (NCT05016063).

### 6.4. HER2 × IL13Rα2 × EphA2 Trivalent CAR T-Cells

Glioblastoma (GBM) is the most common primary malignant brain tumor and is currently incurable. CAR T-cell therapy has shown promising in the treatment of GBM. Bielamowicz et al. [149] demonstrated that targeting three antigens of human epidermal growth factor receptor 2 (HER2), interleukin-13 receptor subunit alpha-2 (IL13Rα2), and ephrin-A2 (EphA2) with a single CAR T-cell product can offset antigen escape, and enhance T-cell effector function. The results showed that combined targeting the antigens of HER2, IL13Rα2, and EphA2 could overcome interpatient variability, with a tendency to capture nearly 100% of tumor cells in the majority of tumors tested in their studies. The CAR T-cells mediated robust immune responses and exhibited improved cytotoxicity and cytokine release than optimal mono-specific and bi-specific CAR T-cells in each patient’s tumor profile. Furthermore, even low doses of CAR T-cells can control patient-derived xenografts (PDXs) established with GBM and improve the survival of treated mice.

## 7. Discussion

As one of the hottest immunotherapy technologies, CAR T-cell therapy has become the pillar therapeutic technology of immunotherapy. The 71% overall response rate (ORR) of CAR T-cell therapy in patients with hematological malignancies was significantly higher than 20% in patients with solid malignancies [40], suggesting that a better understanding of these issues and further development of CAR T-cell therapy are needed.

CAR T-cells can target antigens on tumor cell membranes only. In contrast, intracellular antigens can be targeted only through natural or artificial T-cell receptors (TCR), which are presented as peptides together with MHC molecules [157]. Thus, the selection of surface antigens for CAR T-cell products in solid tumors is challenging with limited antigenic expression in normal tissues but sufficient to induce cytotoxicity to tumors [158], given the lack of tumor-specific antigens thus far. In addition, one of the barriers to the effectiveness of CAR T-cell therapy in solid tumors is antigenic heterogeneity, which weakens the detection of cancer cells by T cells and reduces the efficacy of CAR T-cell therapy.

In fact, many CAR T-cells effectively eliminate tumor cells expressing high levels of the target antigen, but not effectively for tumor or normal cells expressing low levels of the target antigen. The antitumor activity and “on-target, off-tumor toxicity” of CAR T-cells are dependent on the ratio of target antigen density on tumor and normal cells [159]. Consequently, it is critical to find more stable expression and specificity of target antigens.

Scientists have been working to optimize CAR T-cell therapy. CAR T cell therapies have greatly evolved over the past years, including numerous attempts to enhance persistence, proliferation, safety, and efficacy. These efforts and creations include five generations of engineered CAR T cells, inducible switches for CAR T cell killing or regulation, and locoregional CAR T cell delivery, etc. However, minimizing off-target and tumor toxicity of CAR T cells remains challenging.

Neoantigens generated by tumor-specific (somatic) mutation on the surface of solid tumor cells, which have been shown in association with patient survival in human solid cancer [160]. They are attractive targets for CAR T-cell therapy since their expressions are restricted to tumor cells. Neoantigens are highly individualized, therefore, neoantigens-based CAR T-cell therapy may need individualizing and requires that the target neoantigens are membrane expressed. A couple of novel epitopes have been identified, for example, EGFR variant III (EGFR^vIII^) is a tumor-specific protein present in 25–30% of newly diagnosed glioblastomas (GBMs), making it a potential option for CAR T-cell therapy [161]. A couple of clinical trials of CAR T-cell therapy targeting EGFR mutations (NCT02209376, NCT01454596) had been conducted [162]. Mucin 1 (MUC1) is an attractive antigen candidate in cholangiocarcinoma (CCA). Anti-MUC1 CAR T-cells demonstrated a significant specific killing activity against CCA cells (both KKU100 and KKU213A cell lines) at an effector to target ratio of 5:1 [163]. Furthermore, claudin 6 (CLDN6) is a cell surface member protein expressed on multiple solid tumor tissues and its expression levels differ by tumor types [164] while the expression is not observed on normal adult tissue [165]. Dr. John Haanen from Netherlands Cancer Institute presented the results of a first-in-human open-label, multicenter clinical trial to evaluate the safety and preliminary efficacy of a CAR T-cell product targeting CLDN6 during 2022 AACR annual meeting (https://www.aacr.org/about-the-aacr/newsroom/news-releases/new-car-t-cell-therapy-for-solid-tumors-was-safe-and-showed-early-efficacy (accessed on 18 November 2022)). According to the preliminary data of phase I/II clinical trial (NCT04503278), the safety profile of new CAR T-cell products is acceptable, with early signs of efficacy as a monotherapy and in combination with mRNA vaccine in patients with solid tumors. Preclinical models showed that a CLDN6-encoding mRNA vaccine (CARVac) in combination with the CAR T-cell therapy favors CAR T-cell expansion and higher persistence in the blood. This, in turn, increase tumor cell killing [166].

Besides the efficacy and side effects (e.g., CRS and ICANS, etc.) of CAR T-cells in the treatment of solid tumors, another issue to consider is the preparation time and cost. It generally takes from 2 weeks to 1 month to complete the preparation of CAR T-cells from peripheral blood lymphocytes obtained from patients. Most of CAR T-cell products are made of autologous T cells. The promise of autologous cell therapy as a personalized medical intervention is enormous. However, the estimated total cost of autologous CAR T-cell therapy products produced using current manufacturing methods is astronomical ($150,000–$475,000 per treatment), making it harder to compete with “off-the-shelf” cell therapies. So, can allogeneic CAR T-cells be used to achieve the same therapeutic goal? Allogeneic CAR T-cells are often made from T cells donated by healthy donors or umbilical cord blood, so that CAR T-cells can be made and cryopreserved in advance and become the off-the-shelf products, ensuring treatment timeline and saving cost.

Donor-derived allogeneic CAR T-cells have several potential advantages over autologous approaches, such as the immediate availability of frozen batches for patient treatment, standardization of the CAR T-cell product, timing for multiple cell modifications, redosing or combination of CAR T-cells to different targets, and cost reduction using an industrial process. Most importantly, the initial overall treatment response rate of most current autologous CAR T-cell therapies can reach around 90%, with a 5-year sustained remission rate of 58% (from DLBCL data). However, the ORR of the patients to allogeneic CAR T-cell therapy was 67%, the rate of PFS at 6 months was 27%, and the OS was 55% [167]. This efficacy discrepancy suggests that there exist challenges in allogeneic CAR T-cells. The efficacy depends on the persistence of CAR T-cells in the body of patients after infusion. Also, the comparison of autologous with allogeneic CAR T-cells is mainly based on the expansion of CAR T-cells in vivo and the cell detectability at a certain time after the CAR T-cell infusion. The persistence of CAR T-cells is critical for later tumor recurrence. Benjamin et al. reported that allogeneic CAR T-cells were detected in 3 of 21 patients with R/R B-ALL treated with allogeneic CAR T-cells after 42 days, and only 1 patient after 120 days [167]. In contract, the median survival time of autologous CAR T-cells using the first marketed model reached 168 days, and a significant number of patients had detectable autologous CAR T-cells even at 20 months [168]. Therefore, autologous CAR T-cells are more advantageous from the perspective of expansion level. In addition, allogeneic CAR T-cell therapy had a 91% probability of CRS, and 14% were grade 3 and 4 adverse events. In addition, neurotoxicity was also observed in 38% of the patients. Thus, allogeneic CAR T-cells have higher and more severe of side effects than those of autologous CAR T-cells [169], suggesting that transplant reactions may also exist. In addition, allogeneic CAR T-cells cause life-threatening graft-versus-host disease (GVHD) and may be rapidly cleared by the host immune system [170]. In conclusion, autologous CAR T-cell therapy has advantages over allogeneic therapy in terms of efficacy, durability, side effects, and treatment burden. Unfortunately, the leukocytes obtained from the patients at the time of preparation are mostly after multiple treatments, which may affect the quality of patients’ own T cells. If it is defective, the efficacy of autologous CAR T-cells may be poor. An idea is that if people could have their own T cells cryopreserved when they are young and healthy, just in case they need them in the future, as that would be a way to save for a rainy day.

In addition, T cell exhaustion limits the efficacy of CAR T-cell therapy [52]. T cell dysfunction associated with T cell exhaustion is a major obstacle to its efficacy, especially in the solid tumors treated with CAR T-cells [41]. Additional suppressive TME, and the inefficient CAR T-cell trafficking into solid tumors also contribute to the low response rate of solid tumor cells to CAR T-cells [171].

For these reasons, the efficacy and safety of CAR T-cells in solid tumors can be improved by identifying appropriate tumor-associated (particularly specific) antigens, modifying the structure of CAR to enhance the efficacy, specificity, and survival of CAR T-cells, and optimizing the targeting of TME in solid tumor (e.g., lung cancer), exploring combination therapies (i.e., combining with immune checkpoint inhibitors, dual CAR T-cells or trivalent CAR T-cells), or establishing natural ligand-receptor-based CAR T-cells.

Novel technologies are under development to construct new CAR T-cell products and improve CAR T-cell therapy efficacy with safety improvement. Atara Biotherapeutics’ 1XX technology uses one rather than three immunoreceptor tyrosine-based activation motifs (iTAMs) in creating CAR T-cells. This new technique may help prevent the differentiation and exhaustion of counterproductive T-cells and enhance the antitumor activity of CAR T-cells. CAR T-ddBCMA is an autologous anti-BCMA CAR T-cell therapy that uses a novel synthetic binding domain, called a D-Domain, instead of a typical scFv binder [172]. Similarly, CAR T-ddBCMA developed at Arcellx Inc. is an autologous CAR T-cell therapy that encodes a novel non-scFv synthetic binding domain-targeting BCMA with a 4-1BB (CD137, TNFRS9) costimulatory motif and CD3ζ T-cell activation domain. This new product is computationally designed to be highly stable and reduce immunogenicity. According to data from a phase 1 trial (NCT04155749), durable responses and 100% ORR was demonstrated in R/R MM patients with deep and durable responses along with poor prognostic factors.

T-Charge is a novel CAR T-cell cell therapy platform developed by Novartis, which makes CAR T-cells expanded primarily in the patients, eliminating the need for prolonged cell culture in vitro. This could reduce the entire “vein to doorway” timeline by at least half and result in more potent drugs with a better ability to self-proliferate in the body. Novartis presented some positive first-in-human data for two products targeting CD19 and BCMA at 2021 ASH annual meeting, including a product candidate called YTB323. However, this autologous approach has several drawbacks in terms of production time, cost, manufacturing delay, and dependence on the functional fitness of the patient’s T cells, often reduced by disease or previous treatment [173].

The research on CAR T-cell in treatment of solid tumors is still in its infancy, and the beneficial results of the preliminary trials have provided a theoretical basis for their application in the subsequent clinical treatment of solid tumors. While some of these techniques are not currently directly used to treat solid tumors, they may be one day. With the continuous innovation of CAR T design concepts and treatment regimen, CAR T-cell therapy is expected to become main approaches of solid tumor treatment. It should be noted that due to the limited space and the materials collected, this review may not be able to exhaustedly summarize all the potential issues and aspects of each issue, and the omissions are unpreventable and apologized.

## 8. Conclusions

CAR T-cell therapy is effective in hematological malignancies. However, more than half of patients will have a relapse. Of note, CAR T-cell therapy has been even more disappointing in solid tumors. This may be attributed to the antigenic heterogeneity in solid tumors, the risk of on-target off-tumor toxicity, T-cell dysfunction associated with T-cell exhaustion, suppressive TME, and inefficient transport of CAR T-cell trafficking into solid tumors as the major obstacles to the efficacy of CAR T-cell therapy in solid tumors.

## 9. Outlooks

Cancer immunotherapy comes in many forms, including targeted antibodies, cancer vaccines, adoptive cell transfer (ACT), oncolytic viruses (using viruses to infect and destroy cancer cells), checkpoint inhibitors, cytokines, and adjuvants. ACT includes CAR T-cell therapy and tumor-infiltrating lymphocyte (TIL) therapy. CAR T-cell therapy does not always work for every patient and every type of cancer, and some types of cancer are associated with potentially severe but manageable side effects. Although scientists have not yet fully grasped the immune system’s cancer-fighting capabilities, immunotherapy has helped prolong and save the lives of many cancer patients. With the development of modern science, immunotherapy has the potential to become more precise, more personalized, and more effective and with less side effects than current cancer treatments.

The development of CAR T-cells especially for the treatment of solid tumors is progressing. A better more powerful and longer-lived T cells could be engineered, re-programmed, and developed with the help of a regularly clustered regularly interspaced short palindromic repeats (CRISPR) tool to accelerate the design of improved T cell therapies and improve treatment of leukemia and solid cancers [174]. The development of CRISPR-based reprogramming of human immune cells has opened the door to the application of reprogrammed cellular therapies to treat cancer.

Telomeres are TTAGGG repeats that are located at chromosome ends, and their length determines cellular lifespan. Recent studies have shown that intercellular transfer of telomeres rescues T cell senescence and promotes long-term immune memory [175]. Based on this finding, scientists may be able to try to lengthen telomeres during T-cell activation, so that the life span of engineered CAR T-cells can be prolonged.

We look forward to new scientific results that will be soon applied to the clinic for the benefit of cancer.

## Figures and Tables

**Figure 1 cancers-14-05983-f001:**
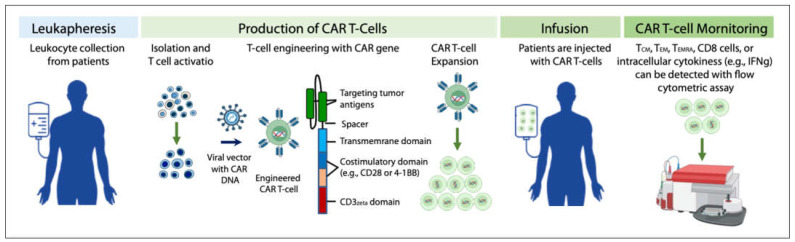
Production, application, and activity monitoring of CAR T-cells.

**Figure 2 cancers-14-05983-f002:**
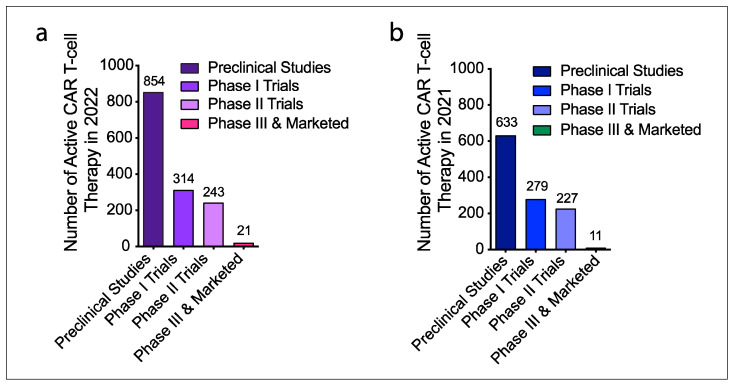
Active CAR T-cell therapies developed in 2022 and 2021. (**a**) A total of 1432 active CAR T-cell therapies have been developed in 2022, including preclinical research (854), phase I (314), phase II (243), phase III and market trials of 21. (**b**) A total of 1150 active CAR T-cell therapies were developed in 2021, including 633 preclinical research, 279 phase I, 227 phase II, 11 phase III and market trials.

**Figure 3 cancers-14-05983-f003:**
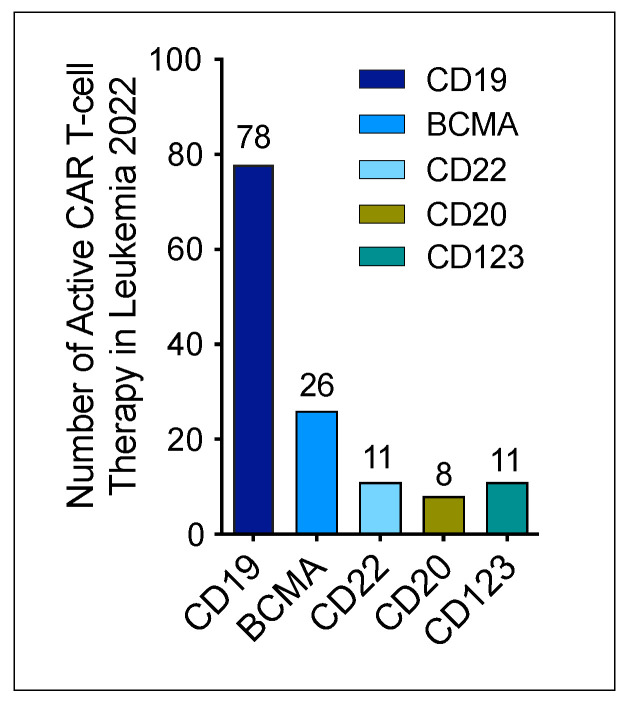
Active CAR T-cell therapies in leukemia in 2022. A total of 134 CAR T-cell trials were conducted in 2022 based on different targets.

**Figure 4 cancers-14-05983-f004:**
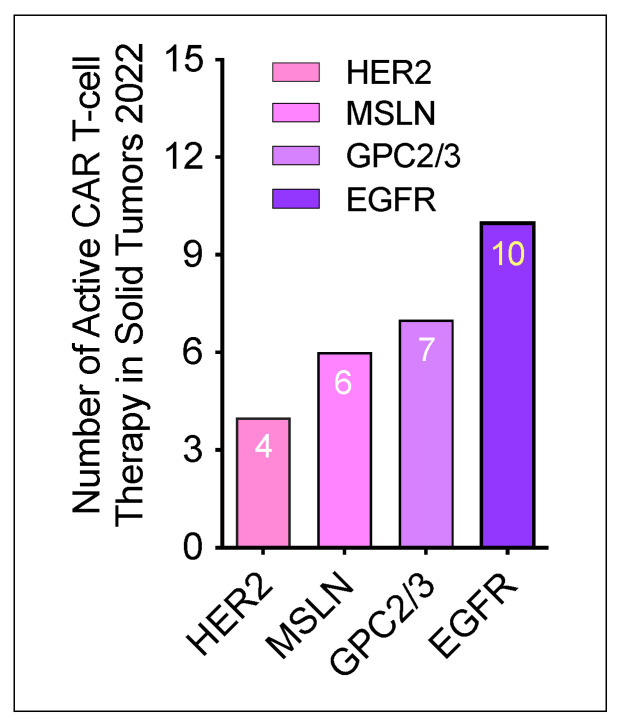
Number of active CAR T-cell trials investigated in 2022. Based on the different antigenic targets, 4 CAR T-cell products are anti-HER2, 6 CAR T-cell products are anti-4 MSLN, 7 CAR T-cell products are anti-4 GPC, 10 CAR T-cell products are anti-EGFR.

**Figure 5 cancers-14-05983-f005:**
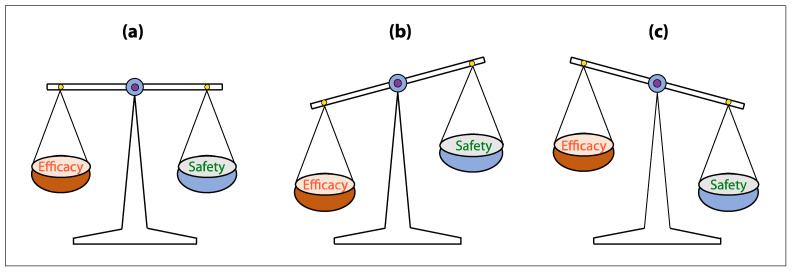
The efficacy and safety of CAR T-cell therapy in the treatment of solid tumors are always at opposite ends of the scale. (**a**) the balance between safety and efficacy, (**b**) higher safety, lower efficacy: safety improves always at the expense of efficacy, (**c**) higher efficacy, lower safety, efficacy improves always at the expense of safety.

**Figure 6 cancers-14-05983-f006:**
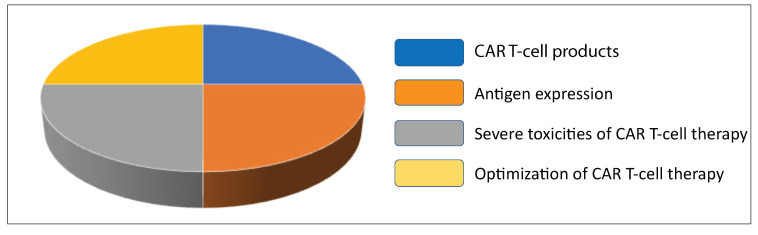
Possible reasons for the failure to sustain remission after CAR T-cell therapies. (1) CAR T-cell products: CAR T-cell product from some patients may not be successfully manufactured due to T cell problems (e.g., autologous T cells obtained from the patients after chemotherapy), or the produced CAR T-cells may not expand adequately either during in vitro culture or after infusion in vivo. Limited persistence of CAR T-cells in other patients is a potential mechanism for disease recurrence. (2). Antigen expression: The absence or downregulation of antigens on the tumor cell surfaces, allows antigen escape as a mechanism of resistance to CAR T-cell therapy. (3). Severe toxicities of CAR T-cell therapy: The fatal toxicity of CAR T-cell therapy (e.g., CRS and/or neurotoxicity) prevents a small percentage of patients benefit from the potential therapeutic of CAR T-cell therapy. (4). Optimization of CAR T-cell therapy in clinical application: For the treatment of the patients with pediatric lymphoma, and solid tumors, CAR T- cell therapy need to be further optimized.

**Table 1 cancers-14-05983-t001:** FDA-Approved CAR T-cell Therapies in hematological malignancies.

Generic Name (Rx)	Brand Name	Target Antigen	Targeted Diseases	Indications (Patient Population)	Manufacturers	Time FDA Approved	Cost of One-Time Infusion
Tisagenlecleucel	Kymriah	CD19	B-cell acute lymphoblastic leukemia (ALL)	Children and young adults with R/R B-cell ALL	Novartis	30 August 2017	USD 475,000
			B-cell non-Hodgkin lymphoma (NHL)	Adults with R/R B cell NHL			
			R/R follicular lymphoma (FL)	Adults with R/R FL		28 May 2022	
Axicabtagene	Yescarta	CD19	B-cell non-Hodgkin lymphoma (NHL)	Adults with R/R B cell NHL	Kite Pharma	18 October 2017	USD 373,000
			Follicular lymphoma (FL)	Adults with R/R FL			
Brexucabtagene	Tecartus	CD19	Mantle cell lymphoma (MCL)	Adults with R/R MCL	Kite Pharma	1 October 2021	USD 373,000
			B-cell acute lymphoblastic leukemia (ALL)	Adults with R/R B-ALL			
Lisocabtagene maraleucel	Breyanzi	CD19	B-cell non-Hodgkin lymphoma (NHL)	Adults with R/R large B cell NHL	Bristol Myers Squibb	5 February 2021	USD 432,055
Axicabtagene ciloleucel	Yescarta	CD19	R/R follicular lymphoma (FL)	Adults with DLBCL	Kite Pharma	5 March 2021 1 April 2022	USD 373,000
Ciltacabtagene autoleucel	Carvykti	BCMA	Multiple Myeloma (MM)	Adults with R/R MM	Janssen Biotech	28 February 2022	USD 465,000
Idecabtagene vicleucel	Abecma				Bristol Myers Squibb	27 March 2021	USD 419,500

Abbreviations: ALL: acute lymphoblastic leukemia; BCMA: B cell maturation antigen; DLBCL, diffuse large B-cell lymphoma; FDA, U.S. Food and Drug Administration; FL, follicular lymphoma; PMBCL, primary mediastinal B-cell lymphoma; R/R, relapsed/refractory. MCL: Mantle cell lymphoma; MM, Multiple Myeloma; NHL: non-Hodgkin lymphoma.

**Table 2 cancers-14-05983-t002:** Most targeted antigens in clinical trials of CAR T-cell therapy in solid tumors.

**Antigen**	**Cancer**	**Phase**	**Identifier (ID)**
EGFR	Lung, colorectal, ovarian, pancreatic, renal cancers	Phase 1/2	NCT01869166
HER2	Central nervous system tumor, pediatric glioma	Phase 1	NCT03500991
EGFR806	Central nervous system tumor, pediatric glioma	Phase 1	NCT03638167
Mesothelin	Ovarian, cervical, pancreatic, lung	Phase 1/2	NCT01583686
PSCA	Lung	Phase 1	NCT03198052
MUC1	Advanced solid tumors, lung	Phase 1/2	NCT03179007, NCT03525782
Claudin 18.2	Advanced solid tumor	Phase 1	NCT03874897
EpCAM	Colon, pancreatic, prostate, gastric, liver	Phase 1/2	NCT03013712
GD2	Brain	Phase 1	NCT04099797
VEGFR2	Melanoma, brain	Phase 1	NCT01218867
AFP	Hepatocellular carcinoma liver cancer	Phase 1	NCT03349255
Nectin4/FAP	Nectin4-positive advanced malignant solid tumor	Phase 1	NCT03932565
CEA	Lung, colorectal, gastric, breast, pancreatic cancer	Phase 1	NCT02349724
Lewis Y	Advanced cancer	Phase 1	NCT03851146
Glypican-3	Liver	Phase 1	NCT02932956
EGFRvIII	Glioblastoma and brain tumor	Phase 1	NCT01454596
IL-13Rα2	Glioblastoma	Phase 1	NCT02208362
CD171	Neuroblastoma	Phase 1	NCT02311621
MUC16 (CA-125)	Ovarian	Phase 1	NCT 02498912
PSMA	Prostate	Phase 1	NCT01140373
AFP	Hepatocellular carcinoma, liver	Phase 1	NCT03349255
AXL	Renal	Phase 1	NCT03393936
CD20	Melanoma	Phase 1	NCT03893019
CD80/86	Lung	Phase 1	NCT03060343
c-MET	Breast, hepatocellular	Phase 1	NCT03060356, NCT03672305
DLL-3	Lung	Phase 1	NCT03392064
DR5	Hepatoma	Phase 1	NCT03638206
EphA2	Glioma	Phase 1	NCT02575261
TAG72	Ovarian	Phase 1	NCT05225363
gp100	Melanoma	Phase 1	NCT03649529
MAGE-A1/3/4	Lung	Phase 1	NCT03356808, NCT03535246
LMP1	Nasopharyngeal	Phase 1	NCT02980315

Abbreviations: EGFR: epidermal growth factor receptor; HER2: human epidermal growth factor receptor 2; PSCA: prostate stem cell antigen; MUC1: mucin 1; EpCAM: epithelial cell adhesion molecule; GD2: disialoganglioside; VEGFR2: vascular endothelial growth factor receptor 2; AFP: alpha fetoprotein; FAP: fibroblast activation protein; CEA: carcinoembryonic antigen; IL-13R: interleukin-13 receptor; CD171: L1 cell adhesion molecule; MUC16: mucin 16; PSMA: prostate-specific membrane antigen; AXL: AXL receptor tyrosine kinase; c-MET: tyrosine-protein kinase Met; DLL-3: delta like canonical notch ligand 3; DR5: death receptor 5; EpHA2: ephrin type-A receptor 2; FR-α: Folate receptor alpha; gp100: glycoprotein 100; MAGE-A: melanoma-associated antigen 3; LMP1: latent membrane protein 1.

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
