# Peer review of "Efficacy, Safety, and Challenges of CAR T-Cells in the Treatment of Solid Tumors"

_cancers, 2022, doi:10.3390/cancers14235983_

Round 1

Reviewer 1 Report

The topic of this review is important, clinically relevant and timely.  However many revisions are needed before this  manuscript can be considered for publication. Specifically,

i)                 the authors should clearly indicate the novel aspects of their review in comparison with the many reviews on the same or similar topic which have been recently published;

ii)                the authors should indicate the criteria they have used to select for their discussion the CAR T cell products used in clinical trials for the treatment of solid tumors; What does major CAR T cell products” mean?

iii)               the authors should combine Section 4 with other related Sections, since Section 4 is redundant;

iv)              Sections 5 should be extensively revised since the mentioned clinical data are derived from the analysis of patients with hematological malignancies and the mechanisms  underlying CAR T cell-related toxicities should be thoroughly and extensively described and discussed;

v)                In Section 7 and 8 the authors should thoroughly discuss  what has been done (i.e. engineering CAR T cells to incorporate inducible switches for CAR T cell killing or modulation, locoregional CAR T cell delivery, etc.) and what can be done to improve CAR T cell safety and efficacy in the treatment of solid tumors.

MINOR COMMENTS:

In Table 1, the list of abbreviations is incomplete.          

Several statements made in the manuscript are misleading  and not supported by the available evidence in the literature. For instance, in page 6 lines 215-216: “… dual targeted CAR T .. associated with disease recurrence.” However, one of the cited references [16] to support this statement describes the use of single targeted CD19 CAR T cells in CLL and not dual targeted CAR T cells. In page 8 line 286, page 12 line 471 and page 19 line 784, the cited references ([40], [69], [131]) are inaccurate.

The authors should provide a footnote with the list of abbreviations used in Table 2. In addition, there are several inaccuracies as some of the clinical trial identifier numbers do not correspond to the CAR T cell products listed.                                

            (i) the ID numbers given for EGFR actually corresponds to that for MUC1

            (ii) the ID number given for EGFR806 does not exist in the clinicaltrial.gov database

            (iii) the ID number given for MUC16 is for CD171

In many parts of the review, the text is poorly written, sometimes with no clear connection between statements (i.e. lines 250-251 and lines 252-254), sometimes with inaccuracies in the statements made (i.e. lines 27-28, lines 179-181, lines 275-276, “hyperproliferation” in line 322) and sometimes lacking the needed follow up statements to clarify the preceding statements made (i.e. lines 707-708). All these  deficiencies make the reading and understanding of the manuscript’s take home message difficult. In addition, several statements made in the manuscript lack proper reference(s) (i.e. lines 90-92, 194-196, 220-222,  315-318, 342-346, 427-429, 553-556, 627-634, 664-669, 847-849). Moreover, the cited references 45-49 are not very informative in terms of the statements made in lines 318-322. The cited reference in line 928 should be added to the list of references.

There are many grammatical inaccuracies and mispellings in the manuscript including some of the figure legends.

Author Response

Editorial Office’s Comments

Please revise the manuscript according to the referees' comments and upload the revised file within 10 days.

Please use the version of your manuscript found at the above link for your revisions.

(I) Please check that all references are relevant to the contents of the manuscript.

     We have carefully checked.

(II) Any revisions to the manuscript should be marked up using the “Track Changes” function if you are using MS Word/LaTeX, such that any changes can be easily viewed by the editors and reviewers.

Track changes was used in the revised manuscript.

(III) Please provide a cover letter to explain, point by point, the details of the revisions to the manuscript and your responses to the referees’ comments.

A cover letter together with the explanation, point by point answers, the details of the revisions to the manuscript and the responses to the referees’ comments has been provided.

(IV) If you found it impossible to address certain comments in the review reports, please include an explanation in your appeal.

We successfully addressed all the questions in the revision.

(V) The revised version will be sent to the editors and reviewers.

It has been submitted to the system.

Comments and Suggestions for Authors

Reviewer 1:

The topic of this review is important, clinically relevant, and timely.  However, many revisions are needed before this manuscript can be considered for publication. Specifically,

  1. the authors should clearly indicate the novel aspects of their review in comparison with the many reviews on the same or similar topic which have been recently published.

Answer: This is an era of immunotherapy, CAR T-cell therapy has exhibited conspicuous clinical efficacy in hematological malignancies. However, the efficacy of CAR T-cell therapy has been disappointing in solid tumors. In this review, we summarized the status of CAR T-cell therapy in solid tumors, together with a summary of CAR T-cell therapy in hematological malignancies, which is a background of CAR T-cell therapy. The novelty and feature of this review is a detailed summary of the status of CAR T-cell therapy in solid tumors, including clinical application, efficacy, safety, and various challenges encountered during translational research and clinical trials. Meanwhile, we also encapsulated not only the potential strategies of improving the efficacy of CAR T-cells and preventing side effects in solid tumors, but also the latest progress of CAR T-cell therapy in solid tumors and outlooks. All these aspects will be popular to the translational researchers (both in academia and industry) and clinicians (Oncologists).

  1. the authors should indicate the criteria they have used to select for their discussion the CAR T cell products used in clinical trials for the treatment of solid tumors; What does “major CAR T cell products” mean?

Answer: The criteria of CAR T-cell products selected for discussion in the limited pages is the attention of the CAR T-cell products and the emergence of clinical accidents in clinical trials. For example, Anti-PSMA CAR T-cell product is one of the pipelines from Tmunity Therapeutics, which was founded by Dr. Carl June, who is one of the pioneers in the field of CAR T-cell research. Tmunity Therapeutics halted the development of its lead CAR T-cell product after the deaths of two patients in a clinical trial in 2021. On this point, it has been described in line 458 in the revised manuscript.

The “major CAR T-cell products” in original line 335 refers to “some main CAR T-cell products, and not all of them”. The word “major” has been corrected to “main” (see it now in line 491).

  • the authors should combine Section 4 with other related Sections since Section 4 is redundant.

Answer: Thank you for the thoughtful suggestion. Following the suggestion, we incorporated the section 4 into other related sections. 

  1. Sections 5 should be extensively revised since the mentioned clinical data are derived from the analysis of patients with hematological malignancies and the mechanisms underlying CAR T cell-related toxicities should be thoroughly and extensively described and discussed.

Answer: We revised the section 5 as 4 now in the revision subtitled “Lesson on safety of CAR T-cells in solid tumors from hematologic malignancies”. We also revised it thoroughly (see the tracking changes in the revised version from line 893 to 1214).

“Experience in safety of CAR-T therapy in hematologic malignancies has accumulated profound lessons, from which physicians may learn to guide their clinical practice in patients with solid tumors, and better manage the safety issues of CAR-T therapy”

Section 6 has also been thoroughly revised.

  1. In Section 7 and 8 the authors should thoroughly discuss what has been done (i.e., engineering CAR T cells to incorporate inducible switches for CAR T cell killing or modulation, locoregional CAR T cell delivery, etc.) and what can be done to improve CAR T cell safety and efficacy in the treatment of solid tumors.

Answer:  Thank you for the suggestions. The section 4 was incorporated into other related sections in the revision. The original section 7 now become section 6, in which, the effectiveness and safety improvement strategies including dual CAR are described, although no dual-target or multi-target CAR T-cell products have been approved for marketing yet. In addition, small paragraphs were added in the discussion section to summarize the development of CAR T-cell products in the past years (line 1878-1883) and indicate the possible limitation of this review (2036-2038).

Scientists have been working to optimize and perfect CAR T-cell therapy. CAR T cell therapies have greatly evolved over the past years, including numerous attempts to enhance persistence, proliferation, safety, and efficacy. These efforts and creations include five generations of engineered CAR T-cells, inducible switches for CAR T-cell killing or regulation, and locoregional CAR T-cell delivery, etc. However, minimizing off-target and tumor toxicity of CAR T cells remains challenging.

It should be noted that due to the limited space and the materials collected, this review may not be able to exhaust all the potential issues and aspects of each issue, and omissions are unpreventable.

MINOR COMMENTS:

In Table 1, the list of abbreviations is incomplete.          

 Answer: Following the suggestion, we made it completely with additions of ALL, BCMA, MCL and NHL (lines 208-210).

Several statements made in the manuscript are misleading and not supported by the available evidence in the literature. For instance, in page 6 lines 215-216: “… dual targeted CAR T .. associated with disease recurrence.” However, one of the cited references [16] to support this statement describes the use of single targeted CD19 CAR T cells in CLL and not dual targeted CAR T cells. In page 8 line 286, page 12 line 471 and page 19 line 784, the cited references ([40], [69], [131]) are inaccurate.

Answer:  The sentence in the original lines 215-216 on page 6 has been deleted in the revision since it is just a plan for following up.

The sentence of “… dual targeted CAR T-cell therapies are still associated with disease recurrence” followed by the (original) references of 15 and 16 are from the review papers, which have been deleted in the revision.

“CAR T-cell therapies targeting either CD19 or CD22 alone have potent antitumor effects, but antigen escape-mediated recurrence frequently occurs. Dual CAR targeting might be applied to overcome this issue” are added in current line 241 with the new references of [32, 33].

The original reference 40 (PMID: 32023374, Liu E, et al. NEJM. 2020, 382(6):545-553) in line 393 has been replaced with a new reference of 46 (PMID: 34666344) in line 394.

For the 5-year survival rate of patients with local staged NSCLC, the original reference 69 (PMID: 33687470. Screening for Lung Cancer: US Preventive Services Task Force Recommendation Statement. JAMA. 2021;325(10):962-97 can be replaced with the ACS data (https://www.cancer.org/cancer/lung-cancer/detection-diagnosis-staging/survival-rates.html). However, this small paragraph has been deleted in the revised version because it is unrelated to MUC1.

The original reference 131 (now 139) is correct. We revised it a word in the sentence as “Bai et al.” Because Melenhorst is the corresponding author, and Bai is the first author.

The authors should provide a footnote with the list of abbreviations used in Table 2. In addition, there are several inaccuracies as some of the clinical trial identifier numbers do not correspond to the CAR T cell products listed.                                

            (i) the ID numbers given for EGFR actually corresponds to that for MUC1:

It has been corrected (table 2).

            (ii) the ID number given for EGFR806 does not exist in the clinicaltrial.gov database

It has been corrected (table 2).

            (iii) the ID number given for MUC16 is for CD17

It has been corrected (table 2).

 Answer: Thank you for the comments. Following the suggestion, we have added the abbreviations under the table 2 in the revised version (lines 554-562). In addition, we have carefully double-checked the ID numbers and corrected the ones of EGFR, EGFR806, MUC16 (i.e., CA-125), and TAG72 for ovarian cancer (see the changes in the revised table 2).

As the anti-CD80/86 CAR T-cells in lung cancer, the identifier of NCT031198052 is correct. However, it was changed/merged (https://clinicaltrials.gov/ct2/history/NCT03198052?V_4=View). Therefore, the trial ID has been replaced with NCT03060343.

In many parts of the review, the text is poorly written, sometimes with no clear connection between statements (i.e. lines 250-251 and lines 252-254),

Answer: We have revised the sentences in the revised version as follows:

“Antigen variants (mutations and/or splicing variants in CD19 gene) -caused escape accounts for 7% to 25% relapse of patients treated with CD19-targeted CAR T-cells due to resistance of CAR T-cells [42]. CD19 CAR T-cell therapy also can lead to deficient or low expression of CD19, which, in turn, result in resistance to the therapy, consequently bringing to DLBCL progression. Spiegel et al. reported that more than 50% of DLBCL patients treated with CD19 CAR T-cells experienced progressive disease because CD19 was absent or low in these patients [39].”

sometimes with inaccuracies in the statements made (i.e., lines 27-28, lines 179-181, lines 275-276, “hyperproliferation” in line 322)

Answer: lines 27-28, revised as “Chimeric antigen receptor (CAR) T-cell therapy is a newly designed adoptive immunotherapy that is able to target and further eliminate cancer cells by engaging with MHC-independent tumor-antigens”

Lines 179-181: It was revised as “BCMA as a target of CAR-T therapy has been approved for the treatment of MM.”

Lines 275-276:  By far, CD19, BCMA, CD22, CD20, CD123, TAA, CD33, CD30, CD38 281 and CS1 have been the favored CAR T-cell therapies for hematological malignancies.

It was revised as Targets of CAR T-cell therapies for hematological malignancies include such as CD19, BCMA, CD22, CD20, CD123, TAA, CD33, CD30, CD38 and CS1” (lines 383-384).

hyperproliferation in Line 322: 

It was revised as “It has been shown that the immunosuppressive tumor microenvironment (TME) provides physical and molecular barriers that prevent T cell infiltration and drive T cell dysfunction and exhaustion in solid tumors [45-49]. Similar mechanisms may exist for the escape of single-target CAR T-cell therapy”. Eventually, this paragraph has been deleted as it is unnecessary.

and sometimes lacking the needed follow up statements to clarify the preceding statements made (i.e., lines 707-708).

Answer: we revised the sentences as CAR T-cell therapy related HLH/MAS has a distinct PET-CT scan from malignancy-related, showing a paradoxical response of hyper-inflammation in CAR-T therapy-related HLH/MAS patients [new citation 131]. Consistently, flow cytometry results showed the expansion of CAR T-cell existing in peripheral blood (PB), and the increased CAR T-cells at different follow-up time points [131]. Anti-IL-6 therapy, steroids, anakinra (a recombinant IL‐1 receptor antagonist) and emapalumab (an anti‐IFNg, approved by the FDA) are recommended for the management of HLH/MAS [127, 131]”

All these deficiencies make the reading and understanding of the manuscript’s take home message difficult. In addition, several statements made in the manuscript lack proper reference (s):

We have added more relevant references in the revised version:

i.e., lines 90-92: A new reference 9 has been added in.

line 194-196 (now 214-216): New references of 27 (PMID: 36153595), and 28 (PMID: 36209106) have been added in.

line 220-222 (now 246-251):

It was revised as “One of the challenges is no ideal antigens existing on AML cells. However, several potential target antigens (e.g., NKG2D ligands, C-type lectin-like molecule-1 (CLL-1), FMA-like tyrosine kinase 3 (FLT3), CD33, and CD23) are under investigation for AML treatment with CAR T-cell therapy (new reference 36, PMID: 35013048). Given that targeted antigens are usually shared between AML cells and myeloid progenitors, switchable CAR-T cells is a key strategy in the construction, thereby increasing safety”.

Line 315-318 (now line 463-465):

The paragraph has been edited thoroughly (line 463-473).

Line 342-346 (now line 566-568)

This sentence was slightly edited and new citations of 52 (PMID: 30843003) and 53 with the PMID of 34298770 published in 2021 in Cancers (MDPI) have been added in (line 566-568).

Line 427-429:  The sentence in original lines of 427-429 has been deleted in the revised version.

Line 553-556 (now 948-951): A new reference of 102 (PMID 31497363) has been added in the revised version.

Line 627-634: This paragraph has been removed in the revised version.

Line 664-669 (now lines 1161-1166): A new reference of 125 (PMID 30181581) has been added in the revised version.

Line 847-849 (now line 1796-1797): A new reference of 150 has been added in the revision.

 Moreover, the cited references 45-49 are not very informative in terms of the statements made in lines 318-322.

Answer:  This paragraph has been deleted, a new paragraph was added in with the references of 50 and 51 in the revised version (line 463-473).

The cited reference in line 928 should be added to the list of references.

Answer:  The cited link has been replaced with a reference of 164 in line 1923.

There are many grammatical inaccuracies and misspellings in the manuscript including some of the figure legends.

Answer: We have carefully edited in the revised version.

Reviewer 2 Report

This is an excellent review of current status of CAR T-cell immunotherapy of solid tumors. The article summaries pros and cons of the fifth pillar of cancer therapy. 

It is important for a wide readership, from those studying medicine and biology, experienced therapists and entrepreneur companies willing to take the risk in entering the market for new products in cancer treatment. It is though sad that just few of the trials mentioned are still in phase 1.  Despite this, I suggest publishing the article as it would be very informative on the current state of the art in this area of research. 

Just a few very minor suggestions for improving the reading of the article.

1. Please add IDN foreshortening into abbreviations, because those outside the field might not know what it stands for. 

2.I wonder whether authors could discuss suggestions on how to prolong the life span of engineered CAR T-cells? For example, recent reports show that by elongating telomeres during T-cell activation, they gain a prolonged potential to divide (APCs transfer via exosomes their telomeres to synapsed T cells during antigen presentation in the draining lymph nodes). Perhaps, a similar procedure during generation of CAR T-cells could be envisaged, i.e. in vitro telomere prolonging phase?

Author Response

-------------------------------------------------------------------------------------------------------

Reviewer 2:

This is an excellent review of current status of CAR T-cell immunotherapy of solid tumors. The article summaries pros and cons of the fifth pillar of cancer therapy. 

It is important for a wide readership, from those studying medicine and biology, experienced therapists and entrepreneur companies willing to take the risk in entering the market for new products in cancer treatment. It is though sad that just few of the trials mentioned are still in phase 1.  Despite this, I suggest publishing the article as it would be very informative on the current state of the art in this area of research. 

Just a few very minor suggestions for improving the reading of the article.

  1. Please add IDN foreshortening into abbreviations, because those outside the field might not know what it stands for. 

AnswerThank you very much for the suggestion, it has been done in the revision. 

  1. I wonder whether authors could discuss suggestions on how to prolong the life span of engineered CAR T-cells? For example, recent reports show that by elongating telomeres during T-cell activation, they gain a prolonged potential to divide (APCs transfer via exosomes their telomeres to synapsed T cells during antigen presentation in the draining lymph nodes). Perhaps, a similar procedure during generation of CAR T-cells could be envisaged, i.e. in vitro telomere prolonging phase?

Answer: Thank you for the suggestion. We have added a paragraph in the section of Outlook, to describe a possible method to prolong the life span of engineered CAR T-cells by lengthening telomeres during T cell activation (line 2073-2077).
